# Healthcare System-to-System Cost Variability in the Care of Pediatric Abdominal Pain-Associated Functional Gastrointestinal Disorders

**DOI:** 10.3390/children8110985

**Published:** 2021-11-01

**Authors:** Michelle Livitz, Alec S. Friesen, Earl F. Glynn, Jennifer V. Schurman, Jennifer M. Colombo, Craig A. Friesen

**Affiliations:** 1Kansas City University of Medicine and Biosciences, 1750 Independence Ave., Kansas City, MO 64106, USA; michelle.a.livitz@gmail.com; 2University of Kansas School of Medicine, 3901 Rainbow Blvd., Kansas City, MO 66160, USA; afriesen17@gmail.com; 3Children’s Mercy Research Institute, 2401 Gillham Rd., Kansas City, MO 64108, USA; efglynn@cmh.edu; 4Division of Pediatric Gastroenterology, Children’s Mercy Kansas City, 2401 Gillham Rd., Kansas City, MO 64108, USA; jschurman@cmh.edu (J.V.S.); jmcolombo@cmh.edu (J.M.C.); 5School of Medicine, University of Missouri Kansas-City, 2411 Holmes Rd., Kansas City, MO 64108, USA

**Keywords:** functional gastrointestinal disorders, irritable bowel syndrome, functional dyspepsia, abdominal pain, abdominal migraine, health care disparity

## Abstract

The purpose of this study was to assess cost variability in the care of abdominal pain-associated functional gastrointestinal disorders (AP-FGIDS) in youth across health systems, races, and specific AP-FGID diagnoses. Patients, aged 8–17 years, with a priority 1 diagnosis corresponding to a Rome IV defined AP-FGID were identified within the Health Facts^®^ database. Total costs were obtained across the continuum of care including outpatient clinics, emergency department, and inpatient or observation units. Cost variability was described comparing different health systems, races, and diagnoses. Thirteen thousand two hundred and fourteen patients were identified accounting for 17,287 encounters. Total costs were available for 38.7% of the encounters. There was considerable variability in costs within and, especially, across health systems. Costs also varied across race, urban vs. rural site of care, and AP-FGID diagnoses. In conclusion, there was considerable variability in the costs for care of AP-FGIDs which is sufficient to support multi-site studies to understand the value of specific tests and treatments. Significant differences in costs by race merit further investigation to understand key drivers.

## 1. Introduction

Pediatric chronic abdominal pain is a common problem with a worldwide prevalence rate of 13.5% [1]. Most will fulfill criteria for an abdominal pain-associated functional gastrointestinal disorder (AP-FGID). There are four AP-FGIDS in youth: irritable bowel syndrome (IBS), functional dyspepsia, abdominal migraine, and functional abdominal pain syndrome [2].

The costs of care of pediatric AP-FGIDs are significant and continue to rise [3,4,5]. These costs are particularly high in patients who are unresponsive or only partially responsive to treatment [6]. In addition, up to 30% of AP-FGIDs persist into adulthood, where the health care costs of IBS alone are over USD 20 billion per year [3,4]. Unfortunately, there are currently no evidence-based pediatric practice guidelines to direct care of these patients. There is a critical need to assess treatment in terms of value or, in other words, the outcomes relative to costs [6,7]. This could be accomplished through multi-site quality improvement projects provided there is sufficient variability in outcomes and/or costs among centers. Currently, there is no process available to assess outcomes across healthcare systems; however, databases are available to assess costs across healthcare systems and AP-FGID diagnoses.

Most previous studies regarding costs have been performed within single sites or utilizing large databases with only emergency department (ED) and/or inpatient data without outpatient clinic cost data. A recent study utilizing a nationally managed healthcare plan database reported total yearly costs (whether directly related to IBS or not) in 1215 pediatric IBS patients across outpatient and inpatient encounters [8]. Data included costs related to IBS as well as costs unrelated to IBS. The Cerner Health Facts^®^ database (Cerner Corporation, North Kansas City, MO, USA) is one of the largest single vendor electronic health record (EHR) databases. It provides multi-site longitudinal data regarding outpatient/clinic, ED, and inpatient/observation encounters, making it an ideal source to assess total costs across the continuum of care for youth with AP-FGIDs.

The primary objectives of the current study were to assess variability in direct costs accrued in the care of AP-FGIDs in youth across healthcare systems, races, ages, gender, and AP-FGID diagnoses.

## 2. Materials and Methods

This study was conducted utilizing a single vendor database, Cerner Health Facts^®^ (Cerner Corporation, North Kansas City, MO, USA). Cerner Corporation is a leading EHR vendor and maintains a publicly available database including data from facilities geographically distributed across the United States. The database contains data from over 68 million patients from 664 facilities within 100 non-affiliated healthcare systems. Health Facts contains data extracted from electronic health records (EHRs) under a data use agreement between Cerner Corporation and individual systems utilizing the Cerner EHR. The database contains data across the continuum of care including outpatient, clinic, emergency department, observation, and inpatient encounters. The data are collected in a longitudinal fashion. Consent and assent were waived by the Institutional Review Board of Children’s Mercy.

### 2.1. Patients

We included all patients meeting an ICD-9 or ICD-10 code corresponding to an AP-FGID, who were between 8 and 17 years of age and who received care from 2010 to 2017. Only encounters with priority 1 AP-FGID diagnoses were utilized. A priority 1 designation indicates that the AP-FGID was the primary reason for the encounter as designated by the care provider. Diagnoses, diagnostic codes, and the % of the total sample for each diagnosis are shown in Table 1.

### 2.2. Variables

In addition to demographic data (i.e., age, gender, and race), we assessed the geographic region and urban/rural status to ensure broad representation in the sample. The study variables included the total charges per encounter across: (1) healthcare systems, (2) races, (3) age groups, (4) urban/rural site of care, and (5) diagnoses. Within Health Facts, the race category also includes the ethnic group “Hispanic”.

### 2.3. Statistical Analysis

Descriptive statistics were assessed for all study variables. Costs were assessed by encounter so that they were not affected by differences in total encounters for different patients. Costs are shown as boxplots reflecting median charges, from the 25th to 75th percentile ranges, and outliers.

## 3. Results

The total final sample was 13,214 patients (mean age 13.3 ± 2.8 years; 62.9% female) with 17,287 encounters. The age and sex distributions are shown in Figure 1. The total encounter numbers by region: Midwest, 7933; South, 3915; West, 2987; Northeast, 2452. Ninety-two percent of encounters occurred in an urban setting. The most common race designations were Caucasian in 81%, African American in 7%, Hispanic in 3%, and Native American in 3%.

Total costs were reported by 46 of the 77 health systems with total charge information available for 38.7% of the encounters overall. There was considerable variation in total costs within and across systems. The distribution of total encounter costs across health systems are shown on a logarithmic scale in Figure 2.

The median costs and cost distributions by race/ethnicity for children (8–12 years of age) and adolescents (13–17 years of age) are shown in Figure 3. While age appeared to have only a small influence, there was variability by race/ethnicity with generally higher costs for Hispanics and lower costs for Native Americans. The relatively high number of consecutive outliers for these two groups suggests that costs may have had a bi-modal or multi-modal distribution which could be obscured in the box plots.

The median costs and cost distributions by race/ethnicity for care delivered in rural locations compared to urban locations is shown in Figure 4. Differences for Hispanics and Native Americans appeared to be more distinct in urban settings. Costs were higher in urban settings compared to rural settings.

The total charges also varied across diagnoses as depicted in Figure 5. There also appeared to be variability within the same AP-FGID diagnosis when identified by ICD-9 codes as opposed to ICD-10 codes. This was particularly true for the diagnosis of functional dyspepsia.

## 4. Discussion

The economics of abdominal pain-associated FGIDs need to be considered in the context of overall health care economics. Globally, healthcare expenditures continue to rise, outpacing gross domestic product (GDP) growth in Europe, North America, and the world’s emerging economies such as the BRICS nations (Brazil, Russia, India, China, and South Africa) [9,10]. Growth in total health expenditures is multi-factorial, driven in part by longer life expectancy and aging populations as well as advances in technology and therapeutics, in addition to public demand [9]. However, it is mostly driven by common health conditions including non-infectious diseases [10]. Increases in healthcare spending and improvement in disease burden have not been equal across socioeconomic groups [11,12].

The economic impact of IBS has been well studied within Europe and North America with some data from Asia and the Middle East, demonstrating high direct and indirect costs including loss of productivity [13,14,15]. In a study of six European countries, in spite of the differences in healthcare systems, direct costs were similar and driven largely by emergency department visits and hospitalizations [7]. Expenditures were similar to those in the United States, although another review demonstrated higher costs in the United States [14]. A study in China found strikingly similar results, with the major drivers of costs being inpatient and outpatient care and loss of productivity accounting for 25% of all costs [15]. IBS accounted for 3.3% of the total healthcare budget in China [15]. While numerous studies have assessed the costs of IBS in adults, there are much less data for children, which prompted the current study. 

In the current study, there was considerable variability between health systems (and to a lesser degree within single systems) with regards to direct costs in the care of AP-FGIDs in youth. Perhaps this is not unexpected, as there are no evidence-based guidelines to direct evaluation and treatment. Variability could be explained, to some degree, by differences in regional algorithms for diagnosis and treatment. Previous studies have reported high variability in costs even within single systems. Dhroove and colleagues evaluated the cost of the diagnostic evaluation in children with chronic abdominal pain and demonstrated a cost range from just over USD 1000 to nearly USD 21,000 within a single institution [4]. Costs were significantly affected by whether an endoscopy was performed. Lane and colleagues evaluated costs within a single pediatric healthcare network and found that referral to a gastroenterologist increased costs five-fold, even in the absence of endoscopy, with significant variability in costs from patient to patient [16]. Among pediatric gastroenterologists, even in the same institution, there was considerable variation in evaluation and treatment [16,17]. Referral to a gastroenterologist does not appear to be driven by symptoms but may be driven by pain perception [16]. While costs are substantially higher in urban settings, this is likely due to the lack of availability of costly tests and endoscopy in rural facilities, with the costs of evaluation being generated after referral to an urban care setting. It is also possible that patients in a rural setting respond better to initial treatment regimens or do not request more expensive tests. It should be noted that the database does not allow for the estimation of indirect costs of loss of productivity and, thus, underestimates total costs. In a previous study in The Netherlands, loss of patient productivity accounted for 22% of overall costs, similar to the indirect costs in adults [3,15].

Costs could also be affected by the patient’s age and/or race/ethnicity. Our data would indicate that age does not appear to drive up costs at an encounter level. We found no differences in encounter costs comparing children and adolescents. We evaluated these as two separate groups, as a previous study found that most children diagnosed with recurrent abdominal pain before the age of 12 years did not have persistent symptoms, while recurrent abdominal pain at the age of 12 years was associated with an increased risk of an AP-FGID at age 16 years [18]. Certainly, age may affect total patient costs as persistent symptoms likely create increased encounters. There also appeared to be wide variability in total direct costs across races, with the highest costs in Hispanic patients and the lowest in Native American patients. Both of these groups had a significant number of outliers making interpretation difficult. Hispanic patients also appeared to be under-represented in the sample. This could be a classification issue, as the database defines Hispanic as a race rather than an ethnicity. Again, at a granular cost level, the reasons for these differences by race are not clear. These could be driven by poorer outcomes, differing sites of care, or social determinants of health. Previously, our group found a significantly increased rate of emergency department care for Hispanic and African American patients (in press). AP-FGIDs are known to be associated with anxiety, depression, stress, and traumatic life events, all of which are factors which affect treatment response [1,19]. Increased costs could also be driven by racial disparities in the identification and treatment of mental health conditions known to affect outcomes, and a poorer response is associated with increased costs [6,20]. This is an important area for future research.

Costs could also be driven up by co-morbid conditions. A previous study of pediatric IBS patients demonstrated higher yearly healthcare costs (not limited to costs directly attributable to IBS) compared to a control group with increased outpatient visits, emergency department visits, and hospitalizations [8]. These patients also had increased co-morbid conditions including mental health conditions, substance abuse, migraines, and other chronic pain conditions which may independently increase costs and impair treatment response in those patients with IBS [8]. We attempted to limit the effect of co-morbid conditions by including only patients with a priority 1 AP-FGID diagnosis; however, co-morbid conditions likely influenced treatment response. Stress and sleep disturbances have been shown to negatively impact outcomes [19]. A previous single-site pediatric study demonstrated a significant increase in costs within partial treatment responders and non-responders [6].

There also appeared to be cost variability across AP-FGID diagnoses. However, interpretation of these costs appeared to be affected by the conversion from ICD-9 to ICD-10. We analyzed ICD-9 and ICD-10 separately, because the transition to ICD-10 occurred shortly before the release of Rome IV, which significantly altered the pediatric criteria for FD and, to a lesser but still substantial degree, for IBS [2]. More importantly, while we utilized specific diagnostic codes, there was significant variability in how physicians utilize and interpret Rome criteria, with only moderate rates of agreement even among pediatric gastroenterologists, making specific AP-FGID diagnoses potentially unreliable [17,21]. While these criteria specifically require symptoms to be present for a minimum of 8 weeks, there is no way to ensure that symptoms were chronic within the database. Lastly, costs could also be affected by misdiagnosis, classifying another disease (e.g., inflammatory bowel disease) as an AP-FGID.

The main weakness is that the database would not capture care delivered at facilities utilizing a different EHR system. A challenge inherent in this and all pediatric studies of chronic abdominal pain is the inconsistencies in diagnosing specific AP-FGIDs. The strength of the current study is that the database allowed for the description of costs across a large population drawn from multiple sites and across the continuum of care with wide geographic distribution [22,23]. Utilizing the Health Facts database allowed for evaluation across centers and across the continuum of care including outpatient, emergency department, and inpatients settings. Unfortunately, while the database provided total encounter costs, costs for specific tests or treatments were only available in 3.5% of patients; consequently, what drove the costs could not be discerned. Importantly, the database does not allow for reliable assessment of treatment outcomes, which may not only affect costs but is vital in understanding value. While there is a need to contain costs, higher short-term costs are not inherently bad if they are associated with better outcomes, which may decrease the total life-time costs and limit morbidity and disability.

## 5. Conclusions

There is sufficient variability in costs related to the evaluation and management of pediatric AP-FGIDs to support multi-site studies to understand the value of specific tests and treatments. An existing pediatric model exists in other diseases such as the ImproveCareNow (ImproveCareNow.org) collaborative study in pediatric inflammatory bowel disease [24]. Such studies could also move the pediatric community towards the development of practice guidelines with evaluation and treatment paradigms that improve outcome while ideally reducing costs. While these conditions are associated with significant costs, enhancing value should be the primary objective.

## Figures and Tables

**Figure 1 children-08-00985-f001:**
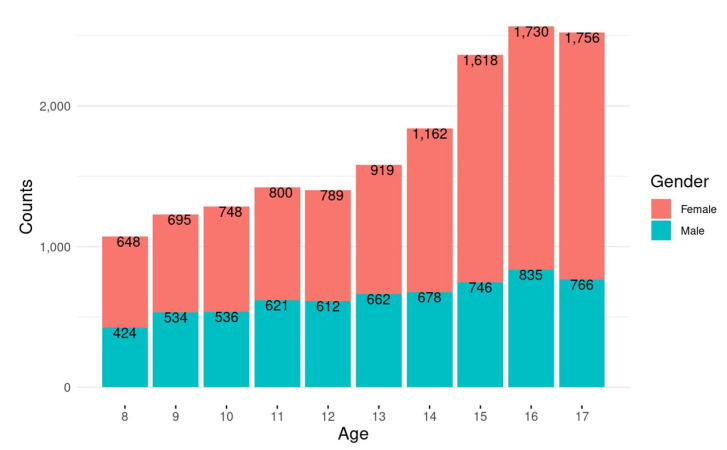
Case distribution by age and gender.

**Figure 2 children-08-00985-f002:**
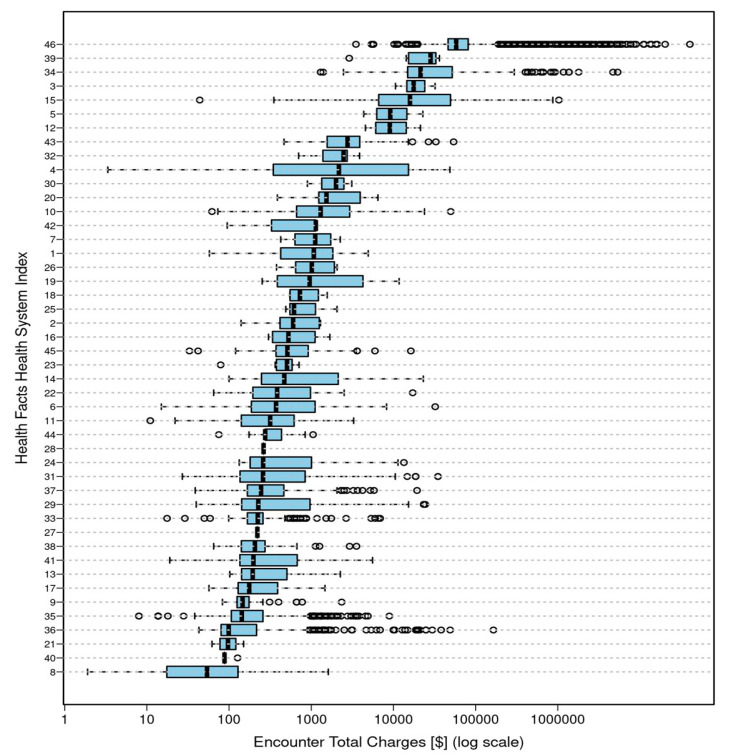
Distribution of total encounter costs across healthcare systems.

**Figure 3 children-08-00985-f003:**
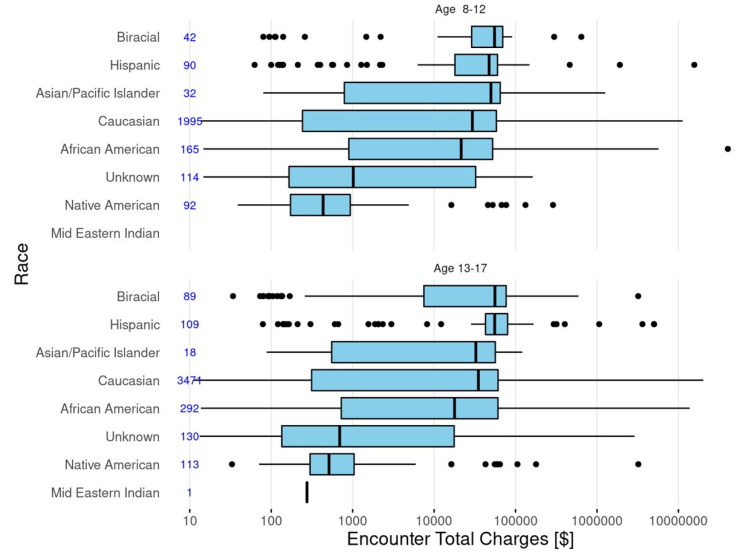
Distribution of total patient charges across races/ethnicity for children (8–12 years of age) and adolescents (13–18 years of age).

**Figure 4 children-08-00985-f004:**
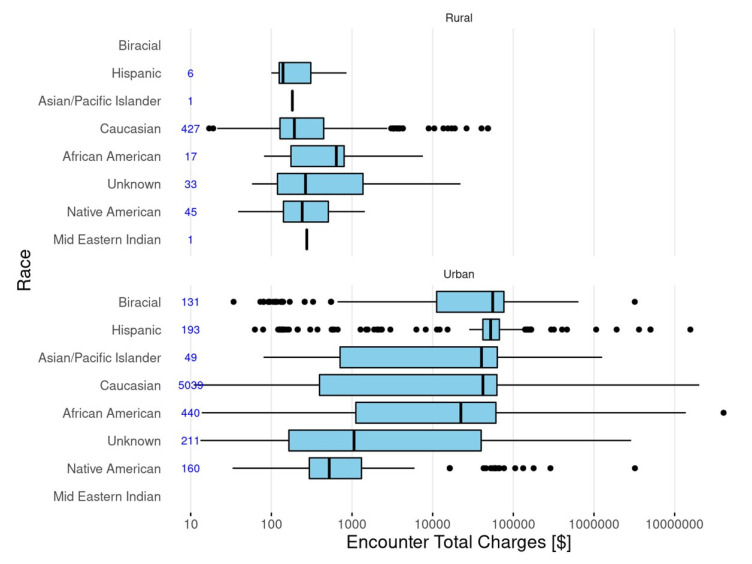
Distribution of total patient charges across races/ethnicity for patients in rural and urban healthcare settings.

**Figure 5 children-08-00985-f005:**
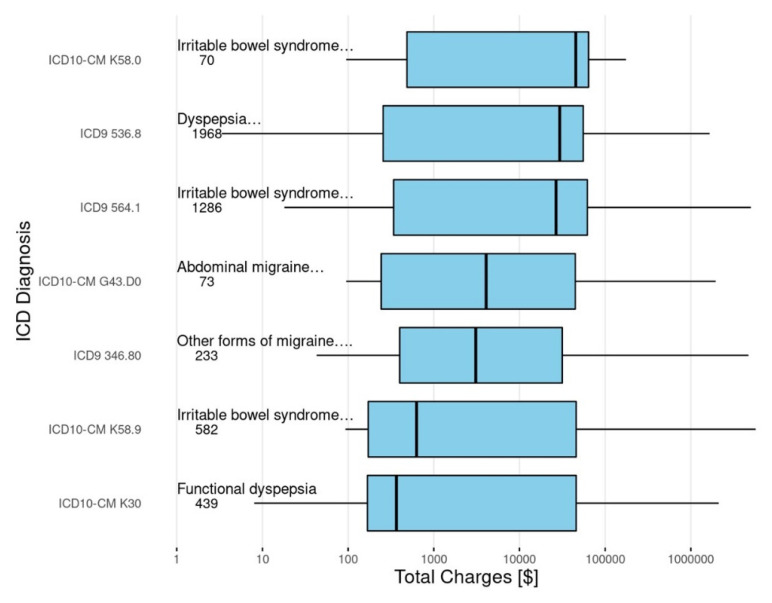
Patient charges across AP-FGID diagnoses.

**Table 1 children-08-00985-t001:** Diagnostic codes utilized to identify various abdominal pain-associated functional gastrointestinal disorders.

Diagnosis	ICD-9 Codes	% of Total	ICD-10 Codes	% of Total
Abdominal Migraine	346.8; 789.06	3.2%	G43.D0; G43.D1	2.3%
Functional Dyspepsia	536.8	37.1%	K30;	13.8%
Irritable Bowel Syndrome	564.1	28.2%	K58.0; K58.1 K58.2; K58.8; K58.9	15.4%

## Data Availability

Data available in Health Facts Database from Cerner Corporation.

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
