# Peer review of "Healthcare System-to-System Cost Variability in the Care of Pediatric Abdominal Pain-Associated Functional Gastrointestinal Disorders"

_children, 2021, doi:10.3390/children8110985_

Round 1

Reviewer 1 Report

Study addresses common questions for pediatricians and pediatric gastroenterologists. The number of ICD codes evaluated in the electronic medical record system is large enough and well distributed geographically to represent a good sample across the U.S.  Overall study addressed many possible data interpretations but i would suggest the following for the authors to consider:

  1. Can authors add more data about health systems included (academic, private clinics, hospital based clinics,...). It would be nice to know cost compressions between tertiary care centers to community hospitals for example.
  2. Is it possible to know if health care costs were different between patients seen by GI specialists vs. patients only managed by primary care providers.
  3. Can authors do a multi-variant analysis to find out statistically any factors associated with increased health care spending. For example median or mean costs would be cut off value to define high cost.
  4. Why authors separated diagnosis via ICD (9 vs 10). The authors in discussion do not give good possible explanation for the difference. It would be nice to mention difference between icd-9 and icd-10 coding and look at co-factors to explain the difference by performing multi-variant analysis.
  5. for racial differences , it would be better to group some minorities in one group and compare to larger ethnic groups. Because some ethnic groups have very small numbers such Hispanics and middle eastern. Would suggest doing Caucasians, AA, and others.

Reviewer 2 Report

Dear Authors,

This submission has been conducted on quite decently large sample size comparing different health systems, races, and 27 diagnoses in a total of 13,214 patients while accounting for 17,287 encounters.

This is a solid evidence to drive some valuable conclusions and fill exisiting gaps in seminal literature.

It should however improve its evidence base.

It is overtly homogeneous relying heavily on US and sources of OECD countries.

Given the fact that Emerging Markets such as BRICS and EM7 are nowdays an economic world engine and the ones contributing ever larger share of morbidity and burden of disease I would warmly recommend introduction of few pieces of evidence in order to increase transnational comparability.

For this purpose I recommend consideration of at least several of the sources listed below alongside few ones at authors own disposal:

https://jamanetwork.com/journals/jamapediatrics/article-abstract/2613463

https://www.ncbi.nlm.nih.gov/pmc/articles/PMC6835015/

https://www.sciencedirect.com/science/article/pii/S0140673617308747 

https://www.mdpi.com/1660-4601/16/17/3043 
